# Incidence of Lower Extremity Amputation in Romania: A Nationwide 5-Year Cohort Study, 2015–2019

**DOI:** 10.3390/medicina59071199

**Published:** 2023-06-25

**Authors:** Emilia Rusu, Horațiu Coman, Andrada Coșoreanu, Ana-Maria Militaru, Horațiu-Cristian Popescu-Vâlceanu, Ileana Teodoru, Doina-Andrada Mihai, Viviana Elian, Norina Alinta Gavan, Gabriela Radulian

**Affiliations:** 1Department of Diabetes, Nutrition and Metabolic Diseases, “Carol Davila” University of Medicine and Pharmacy, Malaxa Clinical Hospital, 030167 Bucharest, Romania; emilia.rusu@umfcd.ro; 2Department of Vascular Surgery, Vascular Surgery Clinic, Cluj County Emergency Hospital, 400347 Cluj-Napoca, Romania; horatiucoman@gmail.com; 3Department of Diabetes, Nutrition and Metabolic Diseases, Malaxa Clinical Hospital, 02441 Bucharest, Romania; ana-maria.militaru@rez.umfcd.ro; 4Department of Diabetes, Nutrition and Metabolic Diseases, 915400 Oltenita, Romania; horatiucristian.ph@gmail.com; 5Department of Diabetes, Nutrition and Metabolic Diseases, “Carol Davila” University of Medicine and Pharmacy, “Prof. Dr. Nicolae Paulescu” National Institute for Diabetes, Nutrition and Metabolic Diseases, 030167 Bucuresti, Romania; ileana.teodoru@umfcd.ro (I.T.); andrada.mihai@umfcd.ro (D.-A.M.); viviana.elian@umfcd.ro (V.E.); gabriela.radulian@umfcd.ro (G.R.); 6Wörwag Pharma Romania SRL, 400267 Cluj-Napoca, Romania; norina.gavan@woerwagpharma.ro

**Keywords:** lower extremity amputations (LEAs), atherosclerosis-related LEAs, diabetes-related LEAs, traumatic LEAs

## Abstract

*Background and Objectives:* The primary objective of this study was to investigate the incidence of lower extremity amputations (LEAs) in a representative population from Romania, in both diabetic and non-diabetic adults, including trauma-related amputations. The secondary objective was to evaluate the trends in LEAs and the overall ratio of major-to-minor amputations. *Material and Methods:* The study was retrospective and included data from the Romanian National Hospital Discharge Records, conducted between 1 January 2015 and 31 December 2019. *Results:* The overall number of cases with LEAs was 88,102, out of which 38,590 were aterosclerosis-related LEAs, 40,499 were diabetes-related LEAs, and 9013 were trauma-related LEAs, with an ascending trend observed annually for each of these categories. Of the total non-traumatic amputations, 51.2% were in patients with diabetes. Most LEAs were in men. The total incidence increased from 80.61/100,000 in 2015 to 98.15/100,000 in 2019. *Conclusions:* Our study reported a 21% increase in total LEAs, 22.01% in non-traumatic LEAs, and 19.65% in trauma-related amputation. The minor-to-major amputation ratio increased over the study period in patients with diabetes. According to these findings, it is estimated that currently, in Romania, there is one diabetes-related amputation every hour and one non-traumatic amputation every 30 min.

## 1. Introduction

Lower limb amputations represent a challenging condition for patients and the healthcare system, causing complex social, professional, educational, and family integration drawbacks by reducing the quality of life and increasing the economic burden.

The two major causes of lower-extremity amputations (LEA) both in low-income and in economically developed countries are diabetes and chronic vascular disease, affecting, predominately, patients between 55 and 75 years old [1]. LEAs are between 15 and 40 times more prevalent in people with diabetes [2].

Diabetic foot is one of the most expensive and severe complications of diabetes. It has been estimated that “every 30 s, a lower limb or part of a lower limb is lost somewhere in the world as a consequence of diabetes” [3]. The latest report of the IDF showed a global increase in the prevalence of diabetes [4]. The prognosis of these patients is unfavorable, with the risk of morbidity and mortality varying with age, with sex, and between different countries; one-year mortality ranges from 10 to 58% [5,6,7]. Following an initial amputation, patients have a high risk of a subsequent intervention, as 19% to 26% of the patients with diabetes require reintervention in the next 12 months and nearly twice as many (45.9%) within the next 24 months [8]; the incidence of contralateral LEAs is about 22% [9], and approximately 37.1% (95% CI 9.4–58.9%) at five years [8].

In addition to this, patients with diabetes have particular features of atherosclerosis: they tend to have severe below-the-knee atherosclerosis with extensive calcifications, with severe involvement of below-the-knee popliteal and tibial arteries, with a high prevalence of long, calcified occlusions [10]. Diabetic neuropathy and peripheral artery disease (PAD) may cause ulcers and infections and increase the risk of lower leg amputation. In addition, trauma is responsible for 5.8% of lower-limb amputations and is more frequent in the second and third decades of life [11]. Cancer accounts for 0.8% of total amputations and is the most prevailing cause in the first and second decades of life [11].

The incidence of amputations varies significantly between different countries, depending on age, genetic and geographic factors, the prevalence of peripheral arterial disease, and diabetes; approximately 90% of the cases involve the lower limb. Moxey et al. showed that the incidence of LEAs in the general population from 1989 to 2010 varied between 5.8 and 31 per 100,000 individuals [5].

In Romania, the prevalence of diabetes is estimated between 8.8–11.6% [12,13] and increases with age. Following this trend, an increase in the absolute number of amputations between 2006–2010 was reported, from 18.1 to 25.2/100,000 in the general population, while in diabetic patients, the rate reached 22.2/100,000 [14].

## 2. Objectives

The primary objective of this study was to investigate the incidence of LEAs between 2015 and 2019 in Romania, in both diabetic and non-diabetic adults, including trauma-related amputations; the secondary objective was to evaluate the trends in LEAs and the overall ratio of major-to-minor amputations.

## 3. Materials and Methods

The study is retrospective and observational, conducted between 1 January 2015 and 31 December 2019, as previously described [14].

Data were obtained from the National School for Public Health, Management, and Health Education, from the Romanian Public Hospital Discharge Records nationwide. The analytic parameters included were age, sex, district, and the primary and secondary diagnosis referring to any amputation procedure for each hospitalization. There were no available data regarding the socio-economic factors, lifestyle behavior (such as smoking or alcohol consumption), or other comorbidities (hypertension, atherosclerotic cardiovascular disease, chronic kidney disease, etc.).

The 10th Revision of the International Classification of Diseases diagnostic code (ICD-10) and the International Classification of Procedures in Medicine codes were used to identify conditions; we identified three diagnostic groups: (1) lower-extremity amputations with atherosclerosis (ATS-related LEAs, ATSR-LEAs); (2) any type of diabetes with lower-extremity amputations at any level (Diabetes-related, DR-LEAs), and (3) traumatic lower-extremity amputations (Trauma-related LEAs, TR-LEAs).

### 3.1. Inclusion and Exclusion Criteria

Only amputated patients were selected for inclusion. The records of patients under 18 years old were excluded.

#### 3.1.1. Definitions

LEAs were categorized as major and minor amputations according to procedure codes.

A major lower extremity amputation was defined as the surgical removal of a part or whole limb proximal to the ankle joint [15]. 

A minor lower extremity amputation was defined as any LEA distal to the ankle joint, with the following ICDs: 10 44338-00 amputation of the toe; 44358 amputation of the toe, including the metatarsal bone; 44364-00 mediotarsal amputation; 44364-01 transmetatarsal amputation; 90557-00 disarticulation of the toe.

Traumatic lower extremity amputations were defined by any trauma-related code of the lower extremity from any diagnosis classification and were analyzed separately.

The diabetes status was classified as no diabetes, type 1 or type 2 diabetes (ICD-10 code E10–E11), and other forms of diabetes (ICD-10 E12–E14), which were recorded in any diagnosis classification. 

The national population was accessed through the National Institute of Public Health statistics, according to the most recent national census from 2011 [16].

Obtaining ethical approval was not required, as the study included anonymous data on the entire population, available publicly.

#### 3.1.2. Statistical Analysis

The categorical variables were presented by the observed frequency (%).

The total number of LEAs was determined each year from 1 January 2015 to 31 December 2019. To calculate the incidence of LEAs, we used the estimated resident population of Romania for each year, according to the Romanian National Institute of Statistics (NIS), on January 1 of each year [17].

We calculated the incidence rate (per 100,000 inhabitants) by dividing the number of cases per year by the corresponding number of people in that population.

For people with diabetes, the incidence of amputations was reported with respect to the general population and the existing data in the medical literature. As there was a significant difference in national statistics as to the exact number of patients with diabetes between 2017 and 2018, we also analyzed the incidence of amputations compared to the number estimated by IDF in 2019 (Table 1).

Data were analyzed as follows: total LEAs in both non-diabetic and diabetic subjects; with type 1 diabetes, type 2 diabetes, or other types of diabetes (OTDM); as well as, separately, each type of LEA. The crude incidence rate of LEAs was stratified by the type of amputation (minor or major), the related causes (ATSR, DR, TR), sex, and age group (there were six different age categories).

In the general population:

Incidence Rate = Patients with LEAs in a given year/All population.

In persons with diabetes, the incidence was reported in 3 different ways:

Incidence rate Globally= Patients with diabetes and LEAs in a given year/All population;

Incidence rate in Diabetes= Patients with diabetes and LEAs in a given year/All patients diagnosed with diabetes;

Incidence rate in Diabetes IDF= Patients with diabetes and LEAs in a given year/All patients diagnosed with diabetes, according to IDF.

All statistical analyses were conducted using Microsoft Excel and the Statistical Package for the Social Sciences (SPSS) 21. A significance level (*p*-value) was not required because the study included the total population, not a representative sample.

## 4. Results

### 4.1. General Characteristics

Over 5 years, between 2015 and 2019, the overall number of cases with LEAs was 88,102, out of which 38,590 were ATS-related LEAs, 40,499 were diabetes-related LEAs, and 9013 were trauma-related LEAs, with an ascending trend observed annually for each of these categories (Table 2). Annual cases of DR- LEA, ATSR-LEA, and TR-LEA, stratified by sex, are illustrated in Figure 1. 

Of the total number of non-traumatic amputations, 51.2% were in patients with diabetes. Most LEAs affected men (Figure 1).

The total incidence increased from 80.61/100,000 in 2015 to 98.15/100,000 in 2019. For non-traumatic amputations (n = 79,089 procedures/5 years), an ascending trend was observed from 72/100,000 in 2015 to 87.85/100,000 in 2019; for ATSR-LEAs, there was an increase from 35.85/100,000 in 2015 to 41.74/100,000 in 2019, as well as for DR-LEAs, where the increase was more significant, from 36.14/100,000 in 2015 to 46.11/100,000 in 2019; similarly, an increase in traumatic amputations was observed, from 8.61/100,000 in 2015 to 10.3/100,000 in 2019 (Table 2). 

### 4.2. LEAs and Sex Distribution

Regarding the sex distribution, LEAs were more predominately found among men, the men to women (M/W) ratio being 2.89/5 years and remaining relatively constant; moreover, the M/W ratio was 2.82/5 years for non-traumatic amputations. The M/W ratio for ATSR- LEAs decreased during the follow-up period, from 3.15 in 2015 to 2.88 in 2019, with an average of 3.05/5 years (Table 2).

Among patients with diabetes, the M/W ratio was lower, at 2.63/5 years, but an increase was observed during the follow-up period, from 2.57 in 2015 to 2.72 in 2019. In addition, this ratio was higher if we refer to traumatic amputations, with a value of 3.61/5 years (3.79 in 2015, reaching 3.87 in 2019) (Table 2).

### 4.3. LEAs and Age Distribution

Not only the total number of cases, but also the numbers of major and minor amputations were consistently higher in the oldest age groups (above 70 years old); in the group of patients under 30 years, only a few cases were observed. However, the increase in the number of cases from both sexes during the next decade of life was significant. Procedures number divided by sex, age groups, and leading causes are shown in Appendix A.

Most non-traumatic LEA cases were observed in the elderly, with approximately 80% occurring in individuals aged ≥60 years (79.63%; n = 62,980), 77.23% representing DR-LEAs (n = 31,279), and 82.15% ATSR-LEAs (n = 31,701) (Appendix A). The incidence of LEAs was three times higher in persons over 60 years old, in the general population, and in patients with diabetes (Appendix A).

The incidence of LEAs had an ascending trend considering the age groups; for patients with diabetes, the maximum incidence was also observed in the category of 60–69 years, but with a very high rate in those over 70 years (15.38/100,000/5 years). In diabetic patients, the incidence in the 60–69 years group was 2.27-fold higher than in the 50–59 years group (Appendix A).

Regarding the ATSR-LEAs, the highest incidence was 18.9/100,000/5 years in the group of patients over 70 years, 3.5 times higher than in the group 50–59 years (5.32/100,000/5 years), and 1.41 times higher compared to the group 60–69 years (13,37/100,000/5 years) (Appendix A).

In the 18–49 years group, a low amputation rate was observed in patients with ATSR-LEAs and in patients with diabetes (Appendix A). 

The highest incidence of Trauma-R amputations was observed in 60–69 years. However, the rate of traumatic amputations in patients under 30 years was ten times higher compared to non-traumatic LEAs (Appendix A).

### 4.4. Diabetes-Related LEAs

The total number of LEAs in patients with diabetes in the observational period was 40,499, of which 28.87% (n = 11,692) were major amputations; 72.43% (n = 29,333) of these cases occurred in men (Table 2).

Reporting the incidence of DR-LEAs to the overall population, the incidence of major amputations in the follow-up period was 11.9/100,000/5 years; there was a general ascending trend of major amputations, increasing from 10.87/100,000 in 2015 to 12.71/100,000 population in 2019 (Table 2). Moreover, greater rates were observed in men, from 14.58/100,000 in 2015 to 17.89/100,000 in 2019. The incidence of major amputations among women with diabetes remained relatively consistent (7.54/100,000/5 years), with the highest incidence being 8/100,000 in 2018 (Appendix A). The M/W ratio for major amputations was 2.09/5 years.

In patients with type 1 diabetes, the incidence of major amputations decreased, both in men (from 1.16/100,000 in 2015 to 0.83/100,000 in 2019) and women (from 0.64/100,000 in 2015 to 0.34/100,000 in 2019). In patients with type 2 diabetes, the trend for major amputations followed an ascending path, both in men (from 5.75/100,000 in 2015 to 7.66/100,000 in 2019) and women (from 2.98/100,000 in 2015 to 3.53/100,000 in 2019). For other types of diabetes, the incidence of major amputations remained relatively consistent in both sexes.

### 4.5. Minor Amputations in Patients with Diabetes

The incidence of minor amputations in patients with diabetes was 29.35/100,000/5 years (n = 28,807), of which 74.38% were in men (n =21,427). The ascending trend was noticed in men, women, and the overall population (Table 2). In men, the incidence increased from 38.69 in 2015 to 50.98 per 100.000 in 2019; in women, the rise occurred from 12.46/100,000 in 2015 to 16.55/100,000 in 2019. The M/W ratio was 2.9/5 years (Appendix A).

### 4.6. Major Amputations in Patients with Diabetes

In patients with diabetes, major amputations were less common than minor amputations, that is, a minor-to-major amputations ratio of 2.46 (2.9 for type 1 diabetes, 2.4 for type 2 diabetes, 2.43 for other types of diabetes; 2.7 in men and 1.95 in women). The minor-to-major amputation ratio increased over the study period in patients with diabetes (Table 3).

Over the study period, the incidence of minor and major amputations increased. On one hand, in patients with type 1 diabetes, the rates of both major and minor amputations decreased, while, on the other hand, in patients with type 2 diabetes, an ascending trend was noticed for both types of amputations.

### 4.7. ATS-Related LEAs

During the follow-up period, 38,590 procedures were registered, of which 75.32% (n = 29,066) occurred in men, with the proportion of major amputations reaching 46.67% (n = 18,009).

The 5-year incidence between 2015 and 2019 of ATSR major LEAs was 18.34/100,000 (27.76/100,000 in men and 9.33/100,000 in women). During this period, there was a global upward trend (from 17.52/100,000 in 2015, reaching the highest rate in 2018, of 19.14/100,000, with a slight decrease to 18.78/100,000 in 2019). In men, an incidence of 26.61/100,000 was attained in 2015, increasing to 29.13/100,000 in 2018, and slightly decreasing to 28.05/100,000 in 2019; in women, the incidence increased from 8.84/100,000 in 2015 to 9.89/100,000 in 2019 (Appendix A). The M/W ratio was 2.84.

The 5-year incidence of minor ATS-related amputations increased during the follow-up period, from 18.33/100,000 in 2015 to 22.96/100,000 in 2019, with a 5-year average of 20.96/100,000. The incidence in men was higher compared to women (32.79/100,000/5 years in men versus 9.65/100.000/5 years in women) with an M/W ratio of 3.25, although the trend was ascending for both genders; in men, there was an increase from 29.1/100,000 in 2015 to 35.26/100,000 in 2019, while, in women, the increase was from 8.06/100,000 in 2015 to 11.17/100,000 in 2019 (Appendix A).

### 4.8. Trauma-Related LEAs

Between 2015 and 2019, 9013 traumatic amputations were registered, with 78.32% (n = 7059) in men and 48.5% (n = 4372) being major amputations (Table 2).

The 5-year incidence of trauma-related LEAs was 9.18/100,000 (14.71/100,000 in men and 3.89/100,000 in women), with an M/W ratio of 3.61. 

A number of 4372 major traumatic amputations were registered, of which 78.84% (n = 3447) were in men, with an M/W ratio of 3.72. The incidence of major amputations had an upward trend in men, from 6.72/100,000 in 2015 to 7.97/100,000 in 2019, and, in women, from 1.51/100,000 to 2.08/100,000, slightly decreasing to 1.81/100,000 in 2019. In addition to this, the overall major traumatic amputation rate increased from 4.06/100,000 in 2015 to 4.83/100,000 people in 2019 (Appendix A).

Regarding the minor amputations, the 5-year incidence was 4.73/100.000 (7.53/100,000 in men and 2.05/100,000 in women), with an M/W ratio of 3.51. After an initial decline from 2015 to 2016, the incidence increased in women and men, from 1.99/100,000 in 2015 to 2.32/100,000 in 2019 and from 7.23/100,000 in 2015 to 8.74/100,000 in 2019, respectively (Appendix A).

## 5. Discussion

Over the past 20 years, the incidence of LEAs decreased worldwide. The findings of our study, which included data between 2015 and 2019, showed an increase in the number of LEAs of all causes (ATSR, DR, TR) in Romania for both sexes. For non-traumatic amputations (79,089 procedures being registered over the five years), an ascending trend was observed, from 72/100,000 in 2015 to 87.85/100,00 in 2019. For ATSR amputations, there was an increase from 35.85 in 2015 to 41.74 per 100,000 population in 2019. For DR-LEAs, the increase was even more significant, from 36.14/100,000 in 2015 to 46.1/100,000 in 2019. Apart from these, there was also an increase in traumatic amputations, from 8.6/100,000 in 2015 to 10.26/100,000 in 2019.

In Romania, 80.61 amputations per 100,000 population were performed in 2005, of which 36.14/100,000 were diabetes-related. Until 2019, the numbers increased steadily, reaching 98.15. Compared to another study that included data from 2006 to 2010, where an average of 4584.4+/−612.42 amputations per year were performed in people with diabetes [14], between 2015 and 2019, our numbers doubled, with performance of 8099.8 amputations/5 years. Moreover, our study reported a 21% increase in total LEAs, 22.01% in non-traumatic LEAs, and 19.65% in trauma-related amputation (11.96% in major LEAs and 28.3% in minor LEAs).

A report evaluating the variations in the number of amputations worldwide included data from 12 countries, most of them European countries, showing that the highest rate for major amputations was reported in countries with low gross domestic product (GDP) and healthcare expenditures, such as Hungary or Slovakia [18]. Mean incidence of major amputations per 100,000 inhabitants reported between 2010 and 2014 was 41.4 in Hungary, 29.1 in Slovakia, and 22.6 in Austria. In our study, the mean incidence of major non-traumatic amputation per 100,000 inhabitants was 30.24, with significant differences between sexes (44.23 in men and 16.87 in women). In Hungary, the incidence of major LEAs between 2004 and 2012 was 42.3/100,000 in the general population and 317.0/100,000 in patients with diabetes [19]. Accordingly, the mean incidence of minor amputations per 100,000 inhabitants between 2010–2014 was 46.7 in Slovakia, 37.7 in Germany, and 33.6 in Austria [18]. In Romania, the incidence of non-traumatic minor amputation per 100,000 inhabitants was 50.31 (477.44 in men and 24.3 in women).

In a more recent study between 2015 and 2019 in Germany, the incidence of major LEAs decreased by 7.3% to 24.2 per 100,000 inhabitants, whereas the incidence of minor LEAs increased by 11.8% to 67.1 per 100,000 inhabitants [20].

In our study, the proportion of DR-LEAs from non-traumatic amputation cases was 51.2%, compared to 45.97% from all-cause LEAs. In Romania, previous reports from 2010 showed that 47.62% of all LEAs were diabetes related, with an incidence of 25.52/100,000 [14]. In our study, the diabetes-related amputations/100,000/year incidence almost doubled. In the VASCUNET Report, the prevalence of diabetes in amputees varied between 25.7 in Finland and 74.3 in Slovakia [18]. The VASCUNET Report also emphasized that patients over 65 had almost five times higher rates of LEAs [18]. Accordingly, our study revealed that patients over 60 had a 3- to 4-fold higher risk of LEAs.

LEAs represent an indicator of the quality of care in the diabetic population [21]. Diabetic patients have a nine-fold higher risk of minor amputation and a five-fold higher risk of major amputations than patients without diabetes. 

In our study, in patients with type 1 diabetes, the incidence of minor and major amputations decreased in both men and women. On the contrary, in patients with type 2 diabetes, the trend was ascending for minor and major amputations in both sexes. In subjects with diabetes, minor LEAs were more common than major LEAs, with a minor-to-major LEAs ratio of 2.46 (2.7 in men and 1.95 in women). The minor-to-major ratio increased over the study period in all types of diabetes. 

A variety of studies in medical literature presented over time the incidence of amputations in diabetic and non-diabetic patients worldwide. Therefore, there are many reasons for incidence variation, such as different definitions for major LEAs, differences in methodology or the denominator population (person level, case-level, procedural level), and various data sources.

Subsequently, a decrease in the incidence of non-traumatic amputations, including subjects with type 2 diabetes, was observed in North American countries, as well as in West European countries, such as Spain, Germany, and Denmark [22,23], as a result of addressing specially designed foot and wound clinics, providing advanced healthcare [24].

In Germany, between 2005 and 2007, the incidence of LEAs in diabetic patients was 176.5/100,000, affecting predominately men (76%). The incidence per 100,000 person/year was higher in the diabetic population versus non-diabetic patients—176.5 (CI 95% 156.0–196.9) versus 20.0 (CI 95% 17.0–23.1) in men, and 76.9 (CI 95% 61.9–91.8) versus 13.4 (CI 95% 10.7–16.2) in women [25]. Another German study conducted between 2005 and 2014 revealed a decline in the incidence rate of amputations in female patients [26].

In Spain, findings from 2001 to 2008 suggested a reduction in the overall number of non-traumatic LEAs and an upward trend in type 2 diabetes patients for minor and major LEAs; the incidence rate in major amputations per 100,000 people per year was 0.48 [24]. Furthermore, in patients with type 1 diabetes, it decreased over the studied period, from 0.59 (95% CI 0.51–0.67) to 0.22 (95% CI 0.17–0.26) per 100,000 people, and increased in subjects with type 2 diabetes, from 7.12 (95% CI 6.84–7.4) to 7.44 (95% CI 7.21–7.73) per 100,000 inhabitants. The incidence rate of minor amputations decreased in type 1 diabetes subjects, from 0.88 (95% CI 0.79–0.98) to 0.43 (95% CI 0.37–0.5) per 100 000 people/year, and increased in type 2 diabetes patients, from 9.23 (95% CI 8.92–9.55) to 10.97 (95% CI 10.66–11.29) per 100,000 people/year. Another study demonstrated that, in type 2 diabetes patients, minor LEAs were more frequent than major amputations (59.72% versus 40.28% [27]. Apart from this, the adjusted incidences of minor and major LEAs were higher in men than women [27]. In men with type 2 diabetes, there was a significant decrease in the incidence of minor and major LEAs from 2004 to 2019 [27].

In England, findings from 1985 to 2005 suggested that amputation rates decreased in patients with type 1 diabetes but increased by 43% in type 2 diabetes patients [28]. Deruaz-Luyet et al. reported a higher incidence rate of LEAs for patients with type 1 diabetes than for type 2 diabetes due to the increased severity of the disease.

In Ireland, a study conducted between 2005 and 2009 revealed a significant increase in LEAs in patients with diabetes, with an incidence that increased from 144 to 175/100,000 [29].

In Denmark, the incidence of major LEAs lowered significantly from 2000 to 2011; however, incidence rates were not entirely different between type 1 diabetes and type 2 diabetes patients [30]. Regarding traumatic amputations, the prevalence between 2010 and 2011 was 4% [31]. In our study, an increase in trauma-related amputations was observed. 

In Sweden, studies showed a decrease in the risk of suffering an amputation, although it was higher in both type 1 and type 2 diabetic patients [32].

In Norway, a 40% reduction in diabetic major lower-limb amputations was observed from 1996 to 2006; a similar reduction was noted in the rate of LEAs as a result of peripheral vascular disease [33].

In the diabetic population of the USA, after a reduction of approximately 43% in the number of LEAs observed between 2000 and 2009, in the following six years, between 2009 and 2015, Geiss et al. reported an increased rate of major and minor amputation by 50%, affecting predominately young adults of 18–64 years [34]. The evolution of diabetes represents a risk factor, and a longer duration of diabetes increases the risk of LEAs [35]. Another study revealed increased rates of non-traumatic LEAs among U.S. Medicare FFS beneficiaries with diabetes aged over 67 years [36]. The highest rates were observed in men, the elderly, and among Black race subjects, while non-traumatic LEAs predominately affected men, young patients, and white or Hispanic adults. Rates of below-the-knee and above-the-knee amputations decreased over the last years, even if, concerning non-traumatic LEAs, numbers increased recently across most states in the USA, with rising rates of toe and foot amputations [36].

Another study in the USA evaluating the incidence of LEAs in type 1 versus type 2 diabetes during 2010–2014 concluded that the incidence of LEAs was higher among patients with type 1 diabetes than patients with type 2 diabetes, 5.79 (5.56–6.03) per 1000 people year versus 1.62 (1.59–1.65) per 1000 people year, while the incidence in the control group of people without diabetes was 0.08 (0.08–0.09) per 1000 people/year [37]. A similar trend was reported in the UK in a study conducted between 2000 and 2008 [15] and in Canada from findings reported between 2012 and 2016 [38].

In Canada, data from 2011–2016 mentioned that diabetes represented the most common cause of amputations, followed by other vascular diseases or infection and trauma (1.3/100,000); the incidence rate among people with diabetes was 280.5 per 100,000 people, with a relative risk of 28.9 [38].

A study in Trinidad and Tobago, over seven years, revealed an increasing amputation rate from 16.5 per 100.000 in 2010 to 38 per 100,000 in 2016, with an average incidence per 7 years of 28 per 100,000 [39].

A study from New Zealand suggested that an increased risk of LEAs was associated with older age at diagnosis of diabetes and with the duration of diabetes [40].

According to previous reports, the 5-year M/W ratio was 2.89 for total LEAs, 3.05 for ATSR-LEAs, 2.63 for DR-LEAs, and 3.61 for TR-LEAs; these results are in concordance with other studies conducted in Germany, Spain, Finland, and the USA [24,25,41,42]. A higher frequency of work-related accidents can explain a greater incidence among men. Other risk factors for atherosclerotic vascular diseases include smoking habits, a higher prevalence of peripheral neuropathy and diabetic foot ulceration, greater BMI or visceral fat, or socioeconomic status; this robust association between LEAs and male gender has been described in many earlier publications [43,44].

In Europe, medical research showed that amputation rates also increased in countries with a high number of active smokers, such as Hungary, Slovakia, and Austria [18]. In our country, in 2015, approximately 30% (28.2%) of the adult population were active smokers (36.7% of men and 28.2% of women) [45]. 

Foot ulcers are the major risk factors for LEAs in diabetic subjects [46]. They precede amputation in 80% of cases [47]. Management of diabetic foot care remains suboptimally controlled, as there is still a need to implement guidelines for the prevention and management of diabetic foot, such as prevention programs to reduce the rate of amputations, early identification of high-risk patients and disease, use of ankle-brachial Index (ABI) test for diagnosis and screening of peripheral artery disease, smoking cessation, medical education, multidisciplinary management of risk factors, and rapid access to specialized foot care clinics. Prevention and early treatment of diabetic foot ulcers can reduce the amputation rate by 50%; thus, establishing a multidisciplinary foot care team is a cornerstone in the management and prevention of diabetic foot ulcers [48].

The decline observed in diabetic LEAs in Germany could be explained by more comprehensive medical care for diabetic foot pathology [49]. Improvements in bypass surgery techniques and new endovascular revascularization techniques are important factors for reducing LEAs’ incidence among patients with PAD [50,51].

A worrying aspect is that more than 50% of the amputations in Romania are performed without any imaging diagnosis or a vascular surgery consultation, which is a situation encountered in other countries. Moxy et al. reported a low prevalence of revascularization of 9% in an unselected nationwide cohort. In Denmark, during 2010–2011, only 6% of patients had undergone revascularization [31]. However, in England, Ahmad et al. found a 30% prevalence of revascularization in an unselected population cohort [52].

The strengths of this study include the use of data that cover almost the whole population of Romania, based on the national records, including all types of LEAs. This allowed us to identify, with a lower risk of bias, the regions with lower reporting rates compared to the national average rate and a need to implement prevention and evaluation programs.

The limitations of our study would be the lack of data regarding the subjects’ socioeconomic status, educational level, smoking status, anthropometric parameters, duration of diabetes, diabetic complications or other comorbidities, or the medications administered. Moreover, there was an assumption that only one amputation was performed during one discharge, leading to an underestimation in the number of LEAs. As well, there might have been a risk of overestimating the incidence of LEAs due to the lack of unique identifiers of one individual, multiple amputations being possible in one patient within a given year.

Lack of accuracy in reporting the type of diabetes using the DRG coding system in the electronic healthcare registers might also be a drawback of this study. Apart from this, the paucity of data regarding the prevalence of diabetes led to reporting the findings in several ways, using the total population, the IDF prevalence, and the national prevalence [53]. 

Our findings have a significant impact. The ascending trend in LEAs can be explained by the aging population, the increase in the prevalence of diabetes between 2015 and 2019, the decline in the number of residents of Romania between these years, the higher proportion of active as well as former smokers, lower GDP per capita, lower healthcare expenditure, poorly controlled diabetes, hypertension, dyslipidemia, poor access to medical healthcare, or the modest number of revascularization procedures. High amputation rates in an area indicate inequalities in access to high-quality diabetes foot care [54].

Moreover, this paper presents new data regarding LEAs in Romania. Persons over 60 years with or without diabetes represent a target population for improving foot care in our country by implementing new prevention, screening, and education strategies. There is still a need for a preventive diabetic foot care strategy, new health and economic policies to avoid fatal outcomes, patient-centered educational programs with an emphasis on the prevention of diabetic foot, and new rehabilitation services.

According to our findings, it is estimated that, currently, in Romania, around 8099.8 amputations are performed yearly in diabetic patients, meaning one diabetes-related amputation every hour and one non-traumatic amputation every 30 min.

## 6. Conclusions

Lower extremity amputations remain a major medical and public health issue in Romania and worldwide; the current study provides valuable information regarding the incidence of LEAs. In our country, the rates of non-traumatic and traumatic LEAs increased over five years in both patients with diabetes and with peripheral vascular disease, which may reflect the national disease burden regarding the lack of foot care programs and the urgent need for implementing such programs to improve the quality of care, prevention, and treatment of PAD and diabetic foot ulcers. 

Considering this, the risk of LEAs increases with age, being more prevalent in older persons, and varies with sex (more frequent in men) and the conditions associated with increased rates in diabetic persons.

Given the heavy emotional implications of the indication of amputation on patients and the financial significance for the health system, practical international approaches to care for various foot conditions still need to be adopted.

## Figures and Tables

**Figure 1 medicina-59-01199-f001:**
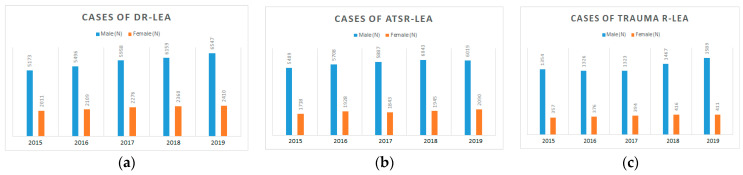
Annual cases of DR- LEA, ATSR-LEA, and TR-LEA, stratified by sex.

**Table 1 medicina-59-01199-t001:** Estimated population with and without diabetes (NIS, IDF).

Years	2015	2016	2017	2018	2019
General population	19,875,542	19,760,585	19,643,949	19,533,481	19,425,873
Diabetic Population	999,192	186,9136	1,785,300	1,161,330	1,704,560
IDF Diabetes	1,278,300	1,278,300	1,278,300	1,278,300	1,278,300

**Table 2 medicina-59-01199-t002:** General characteristics of cases with LEAs in Romania 2015–2019.

	2015	2016	2017	2018	2019	Total
Number of cases with LEA (N)	16,022	16,943	17,681	18,390	19,066	88,102
Men N (%)	11,936 (74.5%)	12,530 (73.95%)	13,168 (74.48%)	13,669 (74.33%)	14,155 (74.24%)	65,458 (74.30%)
Underlying disease N (%)						
DM	7184 (44.84%)	7605 (44.89%)	8234 (46.57%)	8519 (46.32%)	8957 (46.98%)	40,499 (45.97%)
ATS	7127 (44.48%)	7636 (45.07%)	7730 (43.72%)	7988 (43.44%)	8109 (42.53%)	38,590 (43.80%)
Trauma	1711 (10.68%)	1702 (10.05%)	1717 (9.71%)	1883 (10.24%)	2000 (10.49%)	9013 (10.23%)
Incidence of LEA per 10^5^ population	80.61	85.74	90.01	94.15	98.15	89.68
DM	36.14	38.49	41.92	43.61	46.11	41.22
ATS	35.85	38.64	39.35	40.89	41.74	39.28
Trauma	8.60	8.61	8.74	9.63	10.29	9.18
Amputation type N (%)						
Minor	9573 (59.75%)	10,254 (60.52%)	10,911 (61.71%)	11,281 (61.34%)	12,010 (62.99%)	54,029 (61.33%)
Major	6449 (40.25%)	6689 (39.48%)	6770 (38.29%)	7109 (38.66%)	7056 (37.01%)	34,073 (38.67%)
Minor-to-major ratio	1.48	1.53	1.61	1.59	1.70	1.59
Amputation type among DR-LEA N (%)						
Minor	5024 (69.93%)	5374 (70.66%)	5865 (71.23%)	6056 (71.09%)	6488 (72.43%)	28,807 (71.13%)
Major	2160 (30.07%)	2231 (29.34%)	2369 (28.77%)	2463 (28.91%)	2469 (27.57%)	11,692 (28.87%)
Incidence of DR-LEA per 10^5^ Population with DM	718.98	406.87	461.21	733.56	525.47	569.22
Minor	502.81	287.51	328.52	521.47	380.63	404.19
Major	216.17	119.36	132.69	212.08	144.85	165.03
Incidence of DR-LEA per 10^5^ general population						
Minor	25.28	27.20	29.86	31.00	33.40	29.32
Major	10.87	11.29	12.06	12.61	12.71	11.90
Incidence of ATSR-LEA per 10^5^ general population						
Minor	18.33	20.46	21.30	21.76	22.96	20.95
Major	17.52	18.18	18.05	19.14	18.78	18.33
Incidence of TR-LEA per 10^5^ general population						
Minor	4.55	4.24	4.39	4.99	5.47	4.72
Major	4.06	4.38	4.35	4.65	4.83	4.45
Male/female ratio among DR-LEA	2.57	2.61	2.62	2.61	2.72	2.63
Minor	2.97	2.90	2.83	2.88	2.95	2.90
Major	1.90	2.06	2.18	2.08	2.21	2.09
Male/female ratio among ATSR-LEA	3.15	2.96	3.19	3.11	2.88	3.05
Minor	3.45	3.16	3.39	3.29	3.03	3.25
Major	2.87	2.76	2.98	2.91	2.72	2.85
Male/female ratio among TR-LEA	3.79	3.53	3.36	3.53	3.87	3.61
Minor	3.46	3.41	3.38	3.69	3.60	3.51
Major	4.23	3.65	3.34	3.37	4.21	3.73
Number of Nontraumatic-LEA N	14,311	15,241	15,964	16,507	17,066	79,089
Minor	8668	9417	10,049	10,306	10,948	49,388
Major	5643	5824	5915	6201	6118	29,701
Incidence of Nontraumatic-LEA per 10^5^ General population	72.00	77.13	81.27	84.51	87.85	80.51
Minor	43.61	47.66	51.16	52.76	56.36	50.27
Major	28.39	29.47	30.11	31.75	31.49	30.23

**Table 3 medicina-59-01199-t003:** Type of DM among DR-LEAs.

	2015	2016	2017	2018	2019	Total
*Type 1 DM*	1307 (18.19%)	1203 (15.82%)	1102 (13.38%)	1021 (11.99%)	973 (10.86%)	5606 (13.84%)
*Minor*	949	871	814	787	746	4167
*Major*	358	332	288	234	227	1439
*Type 2 DM*	5638 (78.48%)	6164 (81.05%)	6883 (83.59%)	7267 (85.30%)	7725 (86.25%)	33,677 (83.16%)
*Minor*	3903	4341	4879	5104	5552	23779
*Major*	1735	1823	2004	2163	2173	9898
*Other DM*	239 (3.33%)	238 (3.13%)	249 (3.02%)	231 (2.71%)	259 (2.89%)	1216 (3%)
*Minor*	172	162	172	165	190	861
*Major*	67	76	77	66	69	355
*Incidence of DR-LEAs per DM population*						
*Type 1 DM*	6.58	6.09	5.61	5.23	5.01	5.71
*Minor*	4.77	4.41	4.14	4.03	3.84	4.24
*Major*	1.80	1.68	1.47	1.20	1.17	1.46
*Type 2 DM*	28.37	31.19	35.04	37.20	39.77	34.28
*Minor*	19.64	21.97	24.84	26.13	28.58	24.23
*Major*	8.73	9.23	10.20	11.07	11.19	10.08
*Other DM*	1.2	1.2	1.27	1.18	1.33	1.24
*Minor*	0.87	0.82	0.88	0.84	0.98	0.88
*Major*	0.34	0.38	0.39	0.34	0.36	0.36
* **Minor/major ratio among DR-LEAs** *						
*Total*	2.33	2.41	2.48	2.46	2.63	2.46
*Female*	1.70	1.89	2.06	1.95	2.14	1.95
*Male*	2.65	2.66	2.67	2.70	2.85	2.71
*Type 1 DM*	2.65	2.62	2.83	3.36	3.29	2.90
*Type 2 DM*	2.25	2.38	2.43	2.36	2.55	2.40
*Other types of DM*	2.57	2.13	2.23	2.50	2.75	2.43

## Data Availability

Not applicable.

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
