# Peer review of "Incidence of Lower Extremity Amputation in Romania: A Nationwide 5-Year Cohort Study, 2015–2019"

_medicina, 2023, doi:10.3390/medicina59071199_

Round 1

Reviewer 1 Report

Line 47: (Lower limb amputations have been challenging during the last years), the needs proper editing to be more meaningful. LL amputations are still challenging in the current time and possibly in the future. Line 49: (low-income and economically developed countries), this combination is paradoxical. I believe that an economically developed country should have a high income. Line 50: (affecting predominately young patients aged between 55 and 75 years old), I don’t agree that the age 75 is a young age. Line 53: (The prognosis of these patients is unsatisfactory), the term “unsatisfactory” is not a good descriptive word. The paragraph from line 53 to line 58 needs reformatting and language review to be clearer. It is describing diabetic foot that is highlighted in the next paragraph.  Line 59: Highlighting “Diabetic foot” is not necessary as the previous paragraph is about diabetic foot. The whole introduction needs to be reorganized, starting with the global impact of diabetes, and then discussing diabetic foot globally and progressing to mention the Romanian experience. Romania is mentioned in 3 paragraphs; the three paragraphs can be compiled. English language editor may be helpful. Line 83 – 85: (Previous data reported in Romania over a 5-year period of time, between 2006–2010, indicated an increase in the absolute number of amputations (above the ankle amputations), as well as an increase in minor amputations); can be abbreviated to major and minor amputations. Please refer to the definition in line 116.   Line 88: The objectives of the study need to be mentioned in a separate section. Line 93: (Design and study population) can be omitted. Line 94 – 96: (The study was retrospective and observational, including data from the Romanian Public Hospital Discharge Records, conducted between the 1st of January 2015 and to 31st of December 2019, as previously described [14].); This can be changed to (This is retrospective and observational study conducted between the 1st of January 2015 and to 31st of December 2019, as previously described [14].). The source of information may be mentioned in a separate sentence. The (Romanian Public Hospital Discharge Records) can be added to the next paragraph. Line 98: (selected over a 5- year period (2015-2019).); This is a repetition. Line 99: (The parameters included in the analysis were: age, gender, district, and the primary and secondary diagnosis for each hospitalization.); There was no mention of amputation which is the main aim of the study. I suggest rewriting (The analytic parameters included age, sex, district etc.). Line 109: (Trauma-related LEAs, TraumaR). TraumaR may be written TR-LEAs. Line 110: (Inclusion and exclusion criteria); should be highlighted. Line 111: (Only amputated adult patients were selected for inclusion.); can be deleted as it is mentioned in the next sentence (The records of patients under 18 years old were excluded.). Line 116 – 119: No need to mention all these ICD numbers.

Author Response

Thank you for your suggestions!

Line 47: (Lower limb amputations have been challenging during the last years), the needs proper editing to be more meaningful. LL amputations are still challenging in the current time and possibly in the future.

We reformulated the phrase.

Lower limb amputations represent a challenging condition for the patients and the healthcare system, causing complex social, professional, educational, and family integration drawbacks by reducing the quality of life and increasing the economic burden.

Line 49: (low-income and economically developed countries), this combination is paradoxical. I believe that an economically developed country should have a high income. Line 50: (affecting predominately young patients aged between 55 and 75 years old), I don’t agree that the age 75 is a young age.

We reformulated the phrase.

The two major causes of lower-extremity amputations (LEA) in low-income and in economically developed countries as well are diabetes and chronic vascular disease, affecting predominately patients aged between 55 and 75 years old [1].

Line 53: (The prognosis of these patients is unsatisfactory), the term “unsatisfactory” is not a good descriptive word. The paragraph from line 53 to line 58 needs reformatting and language review to be clearer. It is describing diabetic foot that is highlighted in the next paragraph.  

We reformulated the phrase.

Diabetic foot is one of the most expensive and severe complications of diabetes. It has been estimated that “every 30 seconds, a lower limb or part of a lower limb is lost somewhere in the world as a consequence of diabetes” [3]. The latest report of IDF showed a global increase in the prevalence of diabetes [4]. The prognosis of these patients is unfavorable, the risk of morbidity and mortality varying with age, sex, and between different countries; one-year mortality ranges from 10 to 58% [5,6,7]. Following an initial amputation, patients have a high risk of a subsequent intervention, as 19% to 26% of the patients with diabetes require reintervention in the next 12 months and nearly twice as many (45.9%) within the next 24 months [8]; the incidence of contralateral LEAs is about 22% [9], and approximately 37.1% (95% CI 9.4-58.9%) at five years [8].

Line 59: Highlighting “Diabetic foot” is not necessary as the previous paragraph is about diabetic foot. The whole introduction needs to be reorganized, starting with the global impact of diabetes, and then discussing diabetic foot globally and progressing to mention the Romanian experience. Romania is mentioned in 3 paragraphs; the three paragraphs can be compiled. English language editor may be helpful. Line 83 – 85: (Previous data reported in Romania over a 5-year period of time, between 2006–2010, indicated an increase in the absolute number of amputations (above the ankle amputations), as well as an increase in minor amputations); can be abbreviated to major and minor amputations. Please refer to the definition in line 116. 

We reformulated the phrases.

In Romania, the prevalence of diabetes is estimated between 8.8-11.6% [12,13], and increases with age. Following this trend, an increase in the absolute number of amputations between 2006–2010 was reported, from 18.1 to 25.2/100,000 in the general population, while in diabetic patients, the rate reached 22.2/100,000 [14].

 Line 88: The objectives of the study need to be mentioned in a separate section.

The objectives were mentioned in a distinct section.

Line 93: (Design and study population) can be omitted.

Design and study population was deleted.

Line 94 – 96: (The study was retrospective and observational, including data from the Romanian Public Hospital Discharge Records, conducted between the 1st of January 2015 and to 31st of December 2019, as previously described [14].); This can be changed to (This is retrospective and observational study conducted between the 1st of January 2015 and to 31st of December 2019, as previously described [14].). The source of information may be mentioned in a separate sentence. The (Romanian Public Hospital Discharge Records) can be added to the next paragraph.

We reformulated the phrase.

The study is retrospective and observational, conducted between the 1st of January 2015 and to 31st of December 2019, as previously described [14].

Data were obtained from the National School for Public Health, Management, and Health Education, from the Romanian Public Hospital Discharge Records nationwide.

 Line 98: (selected over a 5- year period (2015-2019).); This is a repetition. This phrase has been deleted.

Line 99: (The parameters included in the analysis were: age, gender, district, and the primary and secondary diagnosis for each hospitalization.); There was no mention of amputation which is the main aim of the study. I suggest rewriting (The analytic parameters included age, sex, district etc.).

The analytic parameters included were: age, sex, district, and the primary and secondary diagnosis referring to any amputation procedure for each hospitalization.

Line 109: (Trauma-related LEAs, TraumaR). TraumaR may be written TR-LEAs.

We made this change.

Line 110: (Inclusion and exclusion criteria); should be highlighted.

Inclusion and exclusion criteria were highlighted.

Line 111: (Only amputated adult patients were selected for inclusion.); can be deleted as it is mentioned in the next sentence (The records of patients under 18 years old were excluded.).

Ok

 Line 116 – 119: No need to mention all these ICD numbers.

This phrase has been deleted.

Reviewer 2 Report

The paper is relevant because It presents a large cohort of individuals from Romania whose statistics have been analyzed regarding lower extremity amputations. The introduction and the methodology are well written and easy to follow. However, the significance of the results is lost because there are too many tables (9) and figures (11) some of which are not relevant, or some of the figures are redundant with the tables. Following are some of the recommendations re the tables and figures:

1. In Table 2, the bottom part which is the distribution by Sex (male and female), can already be removed since it is also in Figure 1.

2. Improve the table titles e.g. Table 3. Number of DR-LEAs Distributed (not Divided) by Sex (not Gender), Age and Types of Amputation

3. There is no Table 6 and 7. From Table 5, the numbering goes to Table 8

4. Tables 3-5 are not even referred to in the write up in the text of the results, nor in the discussion. 

5. The re-analysis using the estimates of diabetes from IDF appears unnecessary since the same number is used from 2015-2019. Such extrapolation does not add anything to the value of the data. This makes Table 9 unnecessary 

6. Figures 4-7 just repeat what are already in the tables so these may be removed.

7. Figure 8-9 are also mislabelled. It is not "ratio" but rather "incidence" by age category

By trimming these Tables and figures, the paper can be made better 

Please correct some of terms used e.g. the illustrations are not "graphics" but "Figures" and in the tables, it should not be "Gender" but "Sex"

Author Response

1.In Table 2, the bottom part which is the distribution by Sex (male and female), can already be removed since it is also in Figure 1.

We deleted those data from Table 2

  1. Improve the table titles e.g. Table 3. Number of DR-LEAs Distributed (not Divided) by Sex (not Gender), Age and Types of Amputation

We changed the title of the table.

  1. There is no Table 6 and 7. From Table 5, the numbering goes to Table 8

We corrected the numbering of the tables. We completed the additional data in the supplementary tables.

  1. Tables 3-5 are not even referred to in the write up in the text of the results, nor in the discussion. 

We completed the additional data in the text. We moved Tables 3-5 to the supplementary files.

  1. The re-analysis using the estimates of diabetes from IDF appears unnecessary since the same number is used from 2015-2019. Such extrapolation does not add anything to the value of the data. This makes Table 9 unnecessary 

We deleted this paragraph from the text.

  1. Figures 4-7 just repeat what are already in the tables so these may be removed.

We deleted Figures 4-7.

  1. Figure 8-9 are also mislabelled. It is not "ratio" but rather "incidence" by age category

We changed the title of the Figures.

  1. By trimming these Tables and figures, the paper can be made better 

We appreciate your review. It's very important that the main content of a document to be clear and concise. Moving tables to a separate file, we simplify the main text and improve the flow of information. However, the supplementary file is easily accessible and contains all data.

  1. Comments on the Quality of English Language

Please correct some of terms used e.g. the illustrations are not "graphics" but "Figures" and in the tables, it should not be "Gender" but "Sex"

We changed the terms.
